# Dysregulation of the Bone Marrow Microenvironment in Pediatric Tumors: The Role of Extracellular Vesicles in Acute Leukemias and Neuroblastoma

**DOI:** 10.3390/ijms26115380

**Published:** 2025-06-04

**Authors:** Giovanna D’Amico, Rita Starace, Martina Della Lastra, Danilo Marimpietri, Erica Dander, Fabio Morandi, Irma Airoldi

**Affiliations:** 1Tettamanti Center, Fondazione IRCCS San Gerardo dei Tintori, 20900 Monza, Italy; giovanna.damico@irccs-sangerardo.it (G.D.); r.starace@campus.unimib.it (R.S.); erica.dander@irccs-sangerardo.it (E.D.); 2Laboratorio di Terapie Cellulari, IRCCS Istituto Giannina Gaslini, 16147 Genova, Italy; martinadellalastra@gaslini.org (M.D.L.); danilomarimpietri@gaslini.org (D.M.); fabiomorandi@gaslini.org (F.M.)

**Keywords:** extracellular vesicles, leukemia, neuroblastoma, bone marrow

## Abstract

The role of extracellular vesicles has been extensively studied in physiological and pathological conditions, and growing evidence has pinpointed them as key players in tumor progression, regulation of the metastatic niche, and modulation of anti-tumor immune responses. Indeed, a dynamic transfer of extracellular vesicles between cancer cells and immunological or non-immunological cells homing in the tumor microenvironment exists, and the balance between their release by cancer cells and by normal cells determines cancer progression. Here, we focused on the role of extracellular vesicles in the dysregulation of the bone marrow environment in pediatric tumors such as acute leukemias and neuroblastomata, whose poor prognosis is strictly related to the involvement of such anatomical site. Acute leukemias arise from bone marrow progenitors, whereas approximately 50% of neuroblastoma patients have bone marrow metastases at diagnosis. Thus, here, we discuss the mechanisms underlying the bone marrow dysregulation in pediatric acute leukemias and neuroblastomata with particular emphasis on the involvement of extracellular vesicles.

## 1. Introduction

Extracellular vesicles (EVs) are small particles delimited by a phospholipid bilayer, oriented similarly to that of the plasma membrane, but enriched in cholesterol, ceramide, sphingomyelin, and phosphatidylserine, that are released in the extracellular space by all cell types, irrespective of their origin. EVs express on the surface tetraspanins, including CD9, CD63, CD81, and integrins, but also acquire surface markers of the cell of origin. They contain metabolites, ions, proteins, lipids, and nucleic acids and may be distinguished into several subtypes that differ in size, morphology, and composition. Although the biological role of EVs has been largely underestimated for a long time, it is now recognized that they are crucially important in the cell-to-cell relationship and cell environment communications even at distant sites via biofluids [1,2,3].

EVs were initially divided into four major subclasses, according to their different dimensions and biogenesis: exosomes, microvesicles, apoptotic bodies, and large oncosomes. Afterwards, the nomenclature of EV was updated following the Minimal Information for Studies of Extracellular Vesicles (MISEV) guidelines in 2018 on the basis of the size of EVs. Thus, microvesicles have also been defined as large EVs and exosomes as small EVs [4]. However, in MISEV 2023, it was established that there is no strict consensus on the upper and lower size cut-off (200 nm). Therefore, this type of nomenclature is “recommended with caution” because the size is related to a specific characterization method [5].

Exosomes are spherical vesicles of 50–150 nm released by exocytosis and presumably originated by endocytosis. Microvesicles have an irregular shape with dimensions of 100–1000 nm and are produced by bubble formation on the cellular membrane. Apoptotic bodies, with a diameter of 100–5000 nm, are released by a rigorous apoptotic program, whereas large oncosomes, with dimensions of 1000–10,000 nm, are shed by cancer cells and contain abnormal/transforming macromolecules and other cargo [6,7]. Recent advances in isolation and analytical methods have led to the identification of an ever-increasing number of EV types [8]. Despite such differences, they share the common properties of reflecting the content of the originating cells [9,10].

The functional activities of EVs on recipient cells are mediated by their interactions with surface molecules, including specific ligands, glycans, or lipids, followed by their uptake through both selective and non-selective mechanisms depending on the expression of surface molecules and/or on their overall negative charge. Furthermore, it has been reported that EVs may function by remaining bound to the cell surface without any internalization [9,10]. Thus, the EV signal occurs by cell surface interactions as well as intracellular molecules and implies wide effects on recipient cells due to their pleiotropic functions.

The role of EVs has been extensively studied in physiological and pathological conditions with particular emphasis on their implication in immune regulation and tumor progression with the aim of exploring their potential use as new disease biomarkers, therapeutic tools, or therapy monitoring, as reviewed in [11]. Indeed, growing evidence highlights tumor-derived EVs as main actors in tumor progression, formation of the metastatic niche, and inhibition of anti-tumor immune responses [12,13,14]. Of note, a dynamic transfer of EVs between cancer cells and immunological or non-immunological cells homing in the tumor microenvironment occurs, and the balance between EVs released by cancer cells and those secreted by normal cells, as well as their cargo, determines cancer progression. Cancer cells generally increase the EV release compared to non-malignant cells.

We focused our review on the crosstalk between cancer cells and the bone marrow (BM) highlighting the crucial role exerted by EVs in the dysregulation of such microenvironment in pediatric acute leukemias, both of myeloid and lymphoid origin, and in neuroblastomata. Such malignancies are characterized by a clear involvement of the BM, since acute leukemias arise from BM progenitors, whereas approximately 50% of neuroblastoma patients have BM metastases at diagnosis and present a very poor prognosis.

## 2. Role of EVs in Immune Modulation and Anti-Tumor Responses

EVs may be released by all cells of the innate and adaptive immunity, exerting a prominent role on immune cell functions by signaling in an autocrine, juxtacrine, paracrine, and endocrine manner [13]. They have pro-inflammatory capacities, mediating the differentiation and polarization of type 1 macrophages (M1) and T lymphocytes by carrying mediators such as the high-mobility group box 1 (HMGB1), heat shock proteins (HSPs), damage-associated molecular patterns (DAMPs), and cytokines that may be carried on the EV membrane or as internal cargo (e.g., tumor necrosis factor and transforming growth factor-β) [14,15,16]. Nonetheless, an anti-inflammatory function has been also reported and associated, for example, with M2-derived EVs that transport high levels of the immunosuppressive cytokine IL-10 [17].

Regulation of adaptive immunity is provided for both T and B lymphocytes through different complex mechanisms that include i) differentiation of T lymphocytes into the thymus and of B cells in the BM by molecules such as sphingosine-1-phosphatase lyase 1 and CD24, respectively [18,19], ii) antigen presentation (e.g., carrying functional peptide MHC class I complexes and co-stimulatory molecules) [20,21], iii) cross-presentation by cross-dressing or EV uptake and processing for indirect presentation [22], iv) immune synapses between both T-dendritic cells (DCs) and T–B lymphocytes [23,24], v) immune regulation mediated by immune-checkpoint molecules, including programmed cell death ligand (PDL)-1 and cytotoxic T lymphocyte antigen (CTLA)-4 [25], ectoenzymes, and cytokines, and vi) modulation of anti-microbial responses [26]. Of note, all these activities may also be regulated by nucleic acids and micro-RNA (miRNA) shuttled by EVs.

Finally, the relationship between EVs derived from tumors and the immune response has been largely studied, and their role in suppressing anti-tumor immune responses by acting on macrophages [27], myeloid-derived suppressor cells (MDSCs) [28], DCs [29], NK, T lymphocytes, and regulatory B cells (Breg) [30,31] is now clearly established. EVs function, for example, as carriers of i) immunosuppressive cytokines such as IL-10, transforming growth factor (TGF-β)1, activating MDSCs, Tregs, and down-regulating NKG2D on NK cells [32,33,34], ii) PDL-1, tumor necrosis factor-related apoptosis-inducing ligand (TRAIL), and FAS-L inducing death in T and NK cells [35], iii) HLA-G and adenosine necessary to generate ectoenzymes CD38, CD39, and CD73 [36,37,38,39], and iv) non-coding RNA and miRNA [14]. These mechanisms are involved in tumor progression, but also in the dysregulation of the BM environment of acute leukemias and neuroblastomata, as reported in the following paragraphs.

## 3. Acute Leukemia (ALL and AML)

Acute lymphoblastic leukemia (ALL) is the most common pediatric cancer that occurs consequently to the arrest of typical lymphoid cell maturation in a specific stage of development, thus manifesting as accumulation of malignant and poorly differentiated lymphoid cells within the BM, peripheral blood, and some extramedullary sites. A slight male predominance has been observed, with a peak of incidence at 1–4 years of age [40,41]. As documented by Namayandeh et al. [42], leukemia accounts for 27% of childhood cancers in the United States, 30% in Ireland and France, and 33% in Germany. ALL is by far the most common type of leukemia, accounting for 15–20% of all leukemia cases, followed by acute myeloid leukemia (AML), which is responsible for only a minority of childhood leukemia cases. The mortality rate of leukemia, especially ALL, in children has declined in Europe, the United States, and Japan thanks to the advancement of therapies. Recent studies have reported 5-year overall survival (OS) rates exceeding 90% for ALL [43] and around 65–70% for AML pediatric patients in high-income countries [44]. Despite therapy advancements, acute leukemia still represents one of the leading causes of mortality in children.

Approximately 80% of ALL is represented by B-ALL, a heterogeneous disease consisting of several distinct genetic subtypes characterized by molecular changes such as aneuploidy, chromosome rearrangements, DNA copy number changes, and sequence mutation [45]. Of the subtypes characterized by translocations, the most common in childhood B-ALL is t(12;21)(p13;q22) encoding *ETV6-RUNX1*, which is typically cryptic on cytogenetic analysis and is associated with a favorable prognosis. By contrast, the t(9;22)(q34;q11.2) translocation results in the formation of the Philadelphia chromosome that encodes *BCR-ABL1* and is found in a subset of childhood ALL that is also associated with unfavorable outcomes, although the prognosis has now been improved with combined chemotherapy and tyrosine kinase inhibition [46]. Genomic analyses, particularly transcriptome sequencing, have identified multiple new subtypes not evident on cytogenetic analysis because of cryptic and/or diverse rearrangements or sequence mutations acting as driver lesions. Of these, it is worth mentioning the translocation of *DUX4*, encoding a double-homeobox transcription factor, to the immunoglobulin heavy-chain locus (*IGH*) that is found in 5–10% of B-ALL or the *ETV6-RUNX1*-like ALL occurring almost exclusively in children (representing ~3% of pediatric ALL) and associated with a relatively favorable prognosis [47].

The relevance of these alterations is highlighted by the finding that they are directly involved in the abnormal proliferation of immature lymphoid cells, leading to embryonic and after-birth tumor initiation [43]. Genetic abnormalities are used as diagnostic, prognostic, and predictive biomarkers that play an important role in earlier disease detection, risk stratification, and treatment [45,48]. In particular, in the last few years, the evaluation of minimal residual disease (MRD) has emerged as having an important prognostic and therapeutic impact, equally to genetic subclassification, for risk stratification [49].

Among the various subtypes, the most common is caused by an abnormal clonal proliferation of B cell progenitors, while in the remaining cases, there is the uncontrolled proliferation of mature B cells [50]. Approximately 20% of ALL originate from T cell progenitors. B-ALL derived from B cell progenitors is defined as a neoplasm of precursor lymphoid cells committed to the B cell lineage, typically composed of small-to-medium-sized blast cells with scant cytoplasm, moderately condensed to dispersed chromatin and inconspicuous nucleoli [51]. Current risk assignment incorporates clinical characteristics (age, sex), laboratory studies (white blood cell count and the presence or absence of leukemia in cerebrospinal fluid), as well as characteristics of leukemic blasts (immunophenotype, cytogenetics, molecular diagnostics for the presence of translocation-encoded fusion transcripts, and response to therapy) [52]. The interaction of leukemic blasts with the BM microenvironment is crucial for the development of pre-leukemic cells and in the transition from pre-leukemia to overt leukemia [53]. Furthermore, BM infiltration and early BM response, in terms of tumor burden reduction, represent predictive factors for a successful treatment [54].

Acute myeloid leukemia (AML) is the second most common hematologic malignancy in children. AML is a clonal disorder arising from the transformation of hematopoietic stem or progenitor cells, leading to the accumulation of malignant cells in the BM and other organs [55]. The prognosis for childhood AML has significantly improved over the past 35 years. In the 1980s, nearly all children diagnosed with AML succumbed to the disease. Today, the five-year survival rate for pediatric AML patients has risen to 65–70%. Despite this progress, the prognosis remains less favorable compared to other childhood cancers [44,56]. Approximately 30% of pediatric AML patients experience relapse, and 5–10% ultimately die from disease-related complications or treatment side effects [57]. Although the pathophysiological mechanisms underlying AML development in children are similar to those in adults, the frequency of certain genetic alterations differs between the two populations. Several distinct genetic subgroups have been identified, including core-binding factor (CBF) AML, KMT2A/11q23 rearrangement AML, normal karyotype AML with somatic mutations, AML with unbalanced cytogenetic abnormalities, nucleoporin (NUP)98 11p15/NUP09 rearrangement AML, and acute promyelocytic leukemia (APL) with the PML–RARA rearrangement [56]. Risk group stratification is a key component of therapy planning and is based on classifying patients into different risk groups according to prognostically significant genetic abnormalities and treatment response, assessed through MRD monitoring [58,59]. The role of the BM microenvironment in AML onset and progression has been clearly described [60].

## 4. Neuroblastoma

Neuroblastoma (NB) is the most widespread extracranial solid tumor in childhood, and 90% of cases are diagnosed before 10 years of age, with an average age of 2 years at diagnosis. NB arises from the sympathetic nervous system, mainly from the adrenal medulla, and appears with a heterogeneous clinical presentation, ranging from asymptomatic tumors to diffuse metastases with systemic manifestations [61]. Such differences impact the outcomes of patients since NB may evolve into spontaneous regression as well as into aggressive metastases and death of the patients. Metastases at diagnosis are present in about 50% of patients and mainly involve the BM, but also the bone and lymph nodes.

Diagnosis is based on radiographic imaging and pathology, laboratory tests associated with histologic and cytogenetic analysis, in particular, the amplification of the MYCN oncogene [62]. It has been recently reported [63] that the majority of tumors (69%) combine chromosomal gains or losses with candidate driver mutations at smaller genomic scales, including focal gene amplifications (e.g., MYCN, CDK4, ALK), structural rearrangements (for example, in TERT and ATRX), large deletions (e.g., CDKN2A and ATRX), small insertions/deletions (as in ATRX and NF1), and somatic single-nucleotide variants such as in ALK and those coding for components of the mitogen-activated protein kinase (MAPK) pathways such as HRAS, KRAS, NRAS, and BRAF. In addition, these additional oncogenic drivers support telomere maintenance and MYCN amplification or rearrangement in the TERT locus, representing molecular predictors of a poor outcome.

Regarding the statistical analysis of incidence and mortality, Nong et al. [64] revealed that, globally, the incidence of neuroblastoma in children in 2021 was 5560 cases with 1977 deaths, and from 1990 to 2021, the incidence increased by 30.26%, mortality—by 20.35%. Due to its poor prognosis, many efforts have been made worldwide to stratify patients into different risk groups in order to minimize the therapy and predict potential chemo-resistance. In accordance with the International Neuroblastoma Risk Group Staging System, NB patients are currently stratified as low-risk, intermediate-risk, and high-risk (HR). Low- and intermediate-risk patients have a good prognosis and reach an overall survival rate higher than 90%. By contrast, HR-NB has a poor prognosis, and long-term survival rates remain below 40% despite aggressive therapies. This group represents half of newly diagnosed NB patients, and when metastatic BM disease is present at diagnosis, approximately 10% survive at relapse [65,66]. Thus, BM is commonly used for molecular quantification of MRD and outcome prediction and is the focus of ongoing clinical and research studies.

Since BM infiltration is a common feature in patients with metastatic disease and is correlated with a worse clinical outcome, a better knowledge of the physiological and pathological BM environment may be important to develop novel tools for diagnosis, prognosis, and treatment.

## 5. The BM Niche in Physiological Condition

The BM niche is a complex and highly specialized microenvironment consisting of different cell types, extracellular components, chemical and physical factors. The BM hosts several hematopoietic and non-hematopoietic subpopulations, such as vascular and endothelial cells, megakaryocytes, macrophages, and mesenchymal stromal cells (MSCs) [67]. Two distinct types of niches have been identified, known as the osteoblastic niche and the vascular niche, which work in concert and are regulated by several cell subsets, of which MSCs represent key elements [68,69,70].

The osteoblastic niche, located in the endosteum, includes various cell types such as osteoblasts, osteoclasts, MSCs, and non-myelinating Schwann cells [71]. Osteoblasts work in concert with osteoclasts to maintain bone homeostasis, support and expand hematopoietic stem cells (HSCs) mainly by the Notch signaling [72], and to induce quiescence in HSCs through the production of molecules such as chemokine (C–X–C motif) ligand (CXCL)12 or angiopoietin-1 [69]. Osteoclasts are crucial for the localization of HSCs within the osteoblastic niche and are involved in the formation and maintenance of the cavities that constitute the osteoblastic niche through the receptor activator of nuclear factor kappa B ligand (RANKL) [73].

MSCs are a rare population of multipotent stem cells first identified as adherent cells present in the BM, which exhibit a fibroblast-like morphology and the ability to differentiate in vitro into mesodermal lineages. They are characterized by the expression of specific surface antigens, such as CD105, CD73, CD90, while they do not express hematopoietic, endothelial, and HLA class II markers [74]. MSCs have broad and potent immunoregulatory and regenerative properties both in vitro and in vivo that support their application in clinical settings [75]. More importantly, several scientific studies have demonstrated that, in cases of leukemia onset as well as of metastatic NB, MSCs can support malignant proliferation, survival, and chemoresistance by direct and indirect mechanisms, including the release of EVs [37,76,77,78].

## 6. BM Environment in Acute Leukemia

In recent years, it has been documented that the leukemic BM niche represents an altered microenvironment that supports malignant cells [53,79]. Cancer cells remodel their niche by generating signals and modifying the function of the surrounding cells within the BM. Such remodeling is a significant event in maintaining and controlling the activity of leukemic stem cells during the early stages of the disease and self-reinforcing the malignant environment [80]. These features impact disease progression at the expense of normal hematopoiesis since the progressive changes in both the cellular composition and function of the microenvironment are incompatible with healthy hematopoiesis [70,81,82]. Chemokines, cytokines, enzymes, and other non-protein mediators have been described as part of the signals released by leukemic cells that are able to alter the surrounding microenvironment. AML blasts, through the release of IFN-γ [83], have been shown to upregulate the expression of IFN-induced genes in MSCs. Moreover, thanks to the release of IL-10, IL-35, TGF-β, and indoleamine 2 3-dioxygenase-1 (IDO-1), AML cells have been reported to induce regulatory T cell (Treg) differentiation [79]. Similarly, the release of the enzyme arginase-2 by AML blasts has been correlated with the inhibition of T cell proliferation and the promotion of an M2-like phenotype in surrounding monocytes, thus impacting the BM immune infiltrate. In addition, arginase-2 was demonstrated to inhibit proliferation and differentiation of murine granulocyte–monocyte progenitors and human CD34+ cells, impairing healthy hematopoiesis [84]. More recently, the aberrant expression of S100A8 in AML cells has been associated with an increased production of reactive oxygen species (ROS), which are able to induce a senescence program in BM MSCs [85]. Concerning the ability of leukemia to shape the vascular niche and the surrounding extracellular matrix, AML cells have been shown to secrete metalloprotease-2 (MMP-2) and MMP-9, responsible, through the destruction of tight junction proteins, for the increase in blood–brain barrier permeability, thus facilitating blast dissemination [86]. In addition, in the context of B-ALL, several reports described the expression in membrane-bound form and the release of MMPs, including MMP-9 and MMP-14, associated with relapse and peripheral infiltration [87]. B-ALL cells have also been shown to secrete inflammatory cytokines such as IL-1α, IL-1β, and tumor necrosis factor (TNF)-α and hematopoietic growth factors [88]. Accordingly, the inflammatory mediators IL-6, IL-1β, and TNF-α were described as upregulated in the BM plasma of B-ALL patients compared to HDs and were shown to induce the production of the pro-leukemic factor activin A by BM MSCs [89].

Concerning other microenvironment modifications, studies have highlighted that acute leukemia cells in BM i) prevent the osteogenic differentiation of MSC, ii) shape the extracellular matrix composition of the BM niche, iii) reduce the number of osteoblasts, and iv) modify the vascular composition by increasing the BM microvessel density and blood vessel content [90,91,92,93]. The latter activity is mediated by the ALL secretion of the vascular endothelial growth factor (VEGF), granulocyte–macrophage colony stimulating factor (GM-CSF), M-CSF, granulocyte-CSF, IL-6, and stem cell factor (SCF). In addition, in AML patient-derived xenograft (PDX) models, vascular leakiness and increased hypoxia have been identified as the key features of BM vessel architecture, with nitric oxide (NO) recognized as the primary mediator of this phenotype [94]. Furthermore, leukemic cells secrete factors that disrupt healthy BM, inducing an imbalance in chemokines (e.g., CXCL9, CXCL10) and cytokine production, such as IL-6, IL-10, and TNF-α, that provide an immunosuppressive and permissive microenvironment for leukemic progression [95].

An additional cell component with an important role in leukemic progression is represented by MSCs that can sustain leukemic blasts not only by cell-to-cell contact, but also through the release of cytokines and chemokines. Several studies demonstrated that ALL modulates the BMMSC secretome by increasing the release of CXCL10, CXCL8, CXCL1, CCL22, CCL2, CXCL11, CCL7, and CXCL2 [96]. In this regard, in vivo studies demonstrated that CCL2 increase is related to leukemia/MSC interaction mediated by an osteopontin-dependent mechanism. Accordingly, independent studies found significantly higher levels of CXCL8 and CCL2 in the BM of B-ALL patients at diagnosis compared to healthy donors. In addition to chemokines, blasts also alter the release of cytokines by MSCs such as TGF-β family members endowed with a pro-tumorigenic role, promoting epithelial-to-mesenchymal transition (EMT), invasion, and metastasis [97]. Another molecule involved in the dysregulation of BM is the bone-morphogenetic protein (BMP)-4 that is highly secreted by ALL MSCs and can induce immunosuppressive DCs and polarize macrophages to an M2 pro-tumoral phenotype [98].

## 7. Role of EVs in BM Dysregulation in Acute Leukemia

Several studies have highlighted the significant role of leukemia-derived EVs in modulating the BM niche, establishing a microenvironment that supports leukemia progression [99,100]. Georgievski et al. [101] demonstrated in T and B-ALL PDX that leukemia-derived EVs can induce BM niche transformation. In particular, fluorescently labeled ALL EVs were demonstrated to target murine hematopoietic stem progenitor cells (HSPCs) in the BM, disturbing their quiescence and maintenance. Indeed, metabolomic analysis revealed that ALL-derived EVs were enriched with cholesterol and other metabolites able to promote mitochondrial function in HSPCs and induce their exhaustion. EVs may also vehiculate molecules involved in angiogenesis (e.g., VEGF and miR-210) and immune regulation. Liu et al. demonstrated in vitro and in vivo in a T-ALL model that leukemia-associated EVs activate protein kinase RNA-like endoplasmic reticulum kinase (PERK)-ATF4-JAG1 in endothelial cells, impacting angiocrine factor expression and inducing vascular niche remodeling [102]. Regarding immune modulation, ALL-derived exosomes isolated from the serum of pediatric B-ALL patients were shown to modulate T cell functions by inducing apoptosis and expression of FOXP3 and Treg-related cytokines (e.g., TGF-β and IL-10) [103]. On the same line, 4-1BB-L-enriched EVs released from AML cells were shown to promote the suppressive activity of Tregs [104].

Furthermore, ALL-derived EVs may be internalized by other leukemic cells, contributing to tumor survival and the induction of an aggressive phenotype. In particular, Haque et al. demonstrated that exosomes derived from B-ALL cells are internalized by other leukemic cells, where they promote proliferation by modulating the balance between pro-apoptotic and pro-survival genes [105]. This effect could be ascribed to miR-181-a upregulation in leukemia-associated exosomes. Notably, exosomes derived from leukemic cells at diagnosis and relapse exhibited a stronger ability to promote leukemic proliferation compared to exosomes from the first and second remission [106]. These observations are in accordance with a recent paper from our group, showing that EVs isolated from a B-ALL cell line are efficiently internalized by other leukemic cells, sustaining their survival under nutritional stress culture conditions. Interestingly, activin A, a cytokine released by MSCs upon interaction with leukemic cells, was able to boost the intra-leukemic EV-mediated crosstalk, further improving B-ALL survival [107]. Specifically, we showed that activin A increases the release of EVs by B-ALL cells and modifies EV-associated miRNA cargo. Among the deregulated miRNAs, miR-491-5p was increased, while miR-1236-3p was downregulated. Of note, both miRNAs have been correlated with cancer progression in several tumor types. Similarly, in the context of AML, miR-1246-containing EVs were shown to promote the survival of leukemia stem cells [108].

Leukemia-derived EVs can alter the biological function of MSCs by delivering several types of miRNAs. In this context, miR-146a-5p-, miR-181b-5p-, and miR-199b-3p-enriched exosomes produced by B-ALL cells were shown to activate pro-inflammatory cytokine production in BM MSCs by toll-like receptor (TLR)8 ligation [109]. This phenomenon contributes to the definition of a pro-leukemic microenvironment, detrimental for normal hematopoiesis. Leukemic cells could reshape the BM niche, also affecting the ability of MSCs to differentiate into the osteogenic lineage. Co-culturing the EVs derived from T-ALL cells with BM MSCs led to a reduction in osteogenic differentiation markers and bone mineralization through a miR-34a-5p/Wingless/Integrated (WNT)1-dependent mechanism [110].

In the context of AML, Huan et al. reported that AML-derived EVs participate in the modulation of residual hematopoiesis, indirectly via dysregulation of the BM stromal cell secretome and directly by modulating the biological properties of HSPCs. Indeed, in vitro experiments demonstrated that AML EVs i) promote the loss of clonogenicity and decrease the CXCR4 expression in HSPCs, ii) contribute to the downregulation of SCF and CXCL12 in stromal cells, and iii) induce IL-8 secretion in stromal cells, thus contributing to the etoposide resistance by a Snail-mediated mechanism [111,112]. In particular, the deregulation of the CXCR4/CXCL12 axis was shown to cause the redistribution of HSPCs to the peripheral circulation and a reduction of the BM hematopoietic function [112].

On the other hand, leukemia cells may also represent the specific targets of EVs released by MSCs. In vitro studies by Lyu et al. showed that MSC EVs are able to induce chemoresistance, proliferation, and invasiveness in AML cells by upregulating the calcium-binding protein S100A4 [77]. In the context of B-ALL, we demonstrated that EVs released by MSCs can be actively internalized by ALL cells (manuscript in preparation). Similarly, a recent paper from Karantanou et al. showed an EV-mediated crosstalk between MSCs and BCR-ABL+ B-ALL cells [113]. In particular, the authors identified a mechanism whereby TNF-α secreted by BCR-ABL1^+^ B-ALL cells created a favorable niche for B-ALL progression. This occurs through the downregulation of PLEKHM1 in MSCs, leading to the increased release of EVs enriched in syntenin and syndecan-1. Upon internalization by B-ALL cells, these EVs enhance the phosphorylation of protein kinase B (AKT) and focal adhesion kinase (FAK), thereby promoting BCR-ABL1^+^ B-ALL cell proliferation, migration, and progression. Moreover, Fei et al. demonstrated that stromal-derived exosomes induce galectin-3 upregulation in ALL target cells, promoting leukemia survival under chemotherapy treatment [114]. The mechanisms involved in the dysregulation of the BM microenvironment driven by EVs are summarized in Figure 1.

## 8. BM Environment in Metastatic NB

Several studies showed that metastatic NB cells display distinctive features from primary tumor cells, with upregulation of immunosuppressive molecules (HLA-G and calprotectin) and downmodulation of tumor suppressor genes (i.e., SIRT6, BBC3/PUMA, and CADM4). Nonetheless, more recently, Fetahu et al. [115] reported that BM metastases have tumor cell phenotypes comparable to the primary site, but marked transcriptional differences as tested by single-cell transcriptomic and epigenomic analysis. The authors unraveled the role of monocytes in NB BM metastasis and identified MIF and midkine as specific NB factors rewiring monocytes, which exhibit M1 and M2 features, marked by activation of pro- and anti-inflammatory programs, and express tumor-promoting factors, reminiscent of tumor-associated macrophages. Moreover, the MIF pathway serves as the main communication with stem cells, myeloid cells, B cells, and plasmacytoid dendritic cells, and to a lesser extent with NK and T cells. Next, they assessed changes in cell type abundances and reported an enrichment in T and NK cells, and a depletion of B and myeloid cells in NB metastases compared to controls. Other groups focused on the role of MSC and NB metastasis starting from the observation that MSCs were found increased in number in these patients [116]. MSCs enhanced NB cell growth and showed a reduced ability to differentiate towards osteoblasts and a higher osteogenic potential compared to healthy BM MSCs. In addition, the osteolytic activity was supported by NB cells expressing high levels of the receptor activator of nuclear factor-kB ligand (RANKL) that directly activates osteoclasts, creating the physical space for NB cell growth [117]. The group of Hochheuser [118] demonstrated that in primary NB patient samples, the number of MSCs was significantly increased in metastatic BM compared with NB-free BM, pointing toward a direct or indirect effect of NB cells on MSCs. The same group recently reviewed the role of MSCs in NB, exploring their crosstalk and the potential therapeutic implications in [116].

Similarly to that reported for acute leukemias, another important mechanism underlying the modulation of the BM in NB, thus impacting tumor progression, is represented by EVs. This issue is described in the following section.

## 9. Role of EVs in the Dysregulation of NB BM

Regarding the cross-talk between NB cells and other cell subsets in the BM through the release of EVs, we and others have greatly improved the knowledge in the field. Marimpietri et al. [36,119] first reported the presence of EVs in NB and highlighted the impact of EV cargo in tumor progression. The majority of proteins in the EV-derived NB originated from the cytoplasm and only marginally from the membrane or the nucleus. Recent data from proteomic analyses [120] performed on EVs collected from the serum of NB patients revealed that CD147, HSP90AB1, SLC44A1, CHGA, ATP6V0A1, LFA-1, and CD62L are closely associated with NB and may distinguish NB patients from other individuals. BSG, HSP90AB1, SLC44A1, and CHGA regulate NB progression, whereas ATP6V0A1, LFA-1, and CD62L are involved in immunological synapse formation and immune regulation. Additional proteomic studies on EVs derived from metastatic NB cell lines identified the EV cargo of molecules involved in cell differentiation and proliferation, cell death, metabolic processes, and defense response. These include cell division cycle-associated protein (CDCA)-3, calcium and integrin-binding protein (CIB)-1, and the nuclear pore complex protein Nup107. Of note, hormones, cytokines, and growth factors were not enriched in NB EVs, whereas miRNA or immunosuppressive molecules were present. We [62,121] reported that EVs isolated from BM plasma samples express adenosinergic ectoenzymes that can modulate the BM infiltration by NB cells. NB-derived EVs display higher levels than healthy controls of surface CD38, CD39, CD73, and CD203a/PC-1 and are mostly characterized by higher enzymatic functions. Indeed, the activity of CD39 (witnessed by the conversion of ATP to ADP and AMP) and of CD73 (demonstrated by the ability to convert AMP to ADO) was higher in EVs from NB patients than in those from controls, whereas the enzymatic functions of CD203a/PC-1 and CD38 were similar. Of note, the analyzed adenosinergic ectoenzymes were highly expressed not only by metastatic NB, but also by BM resident cells, including monocytes, granulocytes, and lymphocytes, thus supporting the concept that BM EVs may derive either from mononuclear cells or NB. However, these findings were paralleled by the observation that NB-derived EVs impair T cell proliferation in vitro, suggesting their potential role in the modulation of anti-tumor immune responses.

More recently, we [37] expanded our studies and demonstrated that the large majority of BM EVs present in NB patients derive from MSCs with a smaller contribution from HSCs, mature leukocytes, and NB cells. Such EVs express high levels of CD56, a marker of immune cell activation, and of the immunomodulatory mediators HLA-G, PD-1, and PD-L1. Functional assays revealed these molecules are involved in i) the enhancement of GM-CSF secretion by activated mononuclear cells, ii) inhibition of IL-6 secretion, iii) modulation of IL-2 and IFN-γ production, and iv) inhibition of T cell proliferation. The role of EVs in the dysregulation of NB BM is shown in Figure 2.

## 10. Cancer Vaccines Based on EVs

Beyond the role of EVs in tumor chemoresistance (mainly due to the horizontal transfer of drug resistance proteins and genetic material from chemotherapy-resistant tumor cells) and their potential use as biomarkers of tumor progression as well as therapy monitoring (e.g., thanks to their release in biological fluids), EVs have gained particular attention as a novel immunotherapeutic tool [2,11,122,123]. Indeed, due to their unique ability to vehiculate and deliver bioactive molecules, associated with their resistance in blood and biological fluids, EVs me be used as cancer vaccines and for drug delivery [124,125]. Two main approaches have been investigated in the context of cancer vaccines: the engineering of EVs derived from tumors (Figure 3A) or, alternatively, from immune cells, mainly represented by DCs (Figure 3B) [29]. Gu et al. [126] initially reported that DCs pulsed with tumor-derived EVs were effective in inducing anti-tumor responses despite their cargo of immunosuppressive molecules. Keeping in mind that tumor-derived EVs represent a privileged system to deliver antigenic material to DCs, they have been engineered to increase their immunogenicity and reduce the immunosuppressive features. For example, loading of TLR-activating microRNAs (i.e., miR-142) [127] or IL-12 on tumor-derived EVs and silencing of TGF-β1 induced greater anti-tumor responses compared to unmanipulated EVs [128,129]. Other studies reported satisfactory anti-leukemia responses using leukemia cell-derived EVs engineered to express high levels of the co-stimulatory molecules CD80 and CD86, or low levels of PDL-1. Such approaches improved the DC maturation and antigen presentation, induced cytotoxic T cell responses, increased the M1 level within the tumor, and reduced regulatory T cells, both in vitro and in animal models [130,131]. Nonetheless, it has been suggested that manipulation of DC-derived EVs could be more promising as a cancer vaccine compared to tumor-derived EVs. This is based on the knowledge that DC-derived EVs are free of immunosuppressive cargo, but vehiculate, among others, i) antigens on MHC class I and II molecules, ii) CD80 and CD86 with co-stimulatory functions, iii) NK cell-activating ligands BAT3 and NKG2DL, iv) TNFα, FAS-L, and TRAIL inducing apoptosis on tumors and activating NK cells, v) IL-15Rα inducing activation and proliferation of NK cells, and vi) HSP-70 and -73 with anti-tumor effects [29]. Of note, DC-derived EVs are smaller than intact DCs and can penetrate different biological barriers, including the blood–brain barrier and the blood–tumor barrier, as well as efficiently reach secondary lymphoid organs. DC-derived EVs are commonly obtained by mature DCs generated from peripheral blood monocytes and may be induced to carry tumor-specific antigens, proteins, or peptides derived from different malignant diseases [132]. In this context, it is worth mentioning the ability of DCs pulsed with MAGE or alpha-fetoprotein to release EVs expressing antigens and used against melanoma and hepatocellular carcinoma, respectively. Such DC-derived EVs were able to induce a strong NK and CD8^+^ T cell activation, IFN-γ production, and inhibition of immunosuppressive cytokines (IL-10 and TGF-β1) and regulatory T cell activities [133].

## 11. Concluding Remarks

The role of tumor-derived EVs in the progression of different tumors has been clearly defined in recent years. EVs have a key function in modifying the BM microenvironment to form the pre-metastatic niche through the increase in vascular permeability, the re-modeling of the extracellular matrix, BM cell recruitment, the increase in angiogenesis, and immunosuppression. These features are related to proteins, miRNA, mRNA, and surface receptors derived from parental tumor cells [134,135]. Here, we summarized data present in the literature regarding the role of EVs in the dysregulation of the BM microenvironment in NB, ALL, and AML. Recent studies have characterized the phenotype of EVs from BM samples of patients with NB and hematopoietic malignancies, and other studies are ongoing (Marimpietri et al., in preparation). These studies will help to understand the mechanism(s) underlying their immunosuppressive activities, their role in tumor progression, and their possible contribution to the failure of immunotherapeutic strategies. Indeed, EVs may sequester antibodies directed against tumor antigens (which are also expressed on the surface of tumor-derived EVs) or prime DCs to generate tolerance against tumors.

In addition, a comprehensive analysis of surface markers and a genomic/proteomic analysis of EVs may lead in the future to the use of BM-derived EVs as novel biomarkers for diagnosis and response to treatment in NB and ALL/AML patients, as already investigated for other human solid and hematological tumors [136].

## Figures and Tables

**Figure 1 ijms-26-05380-f001:**
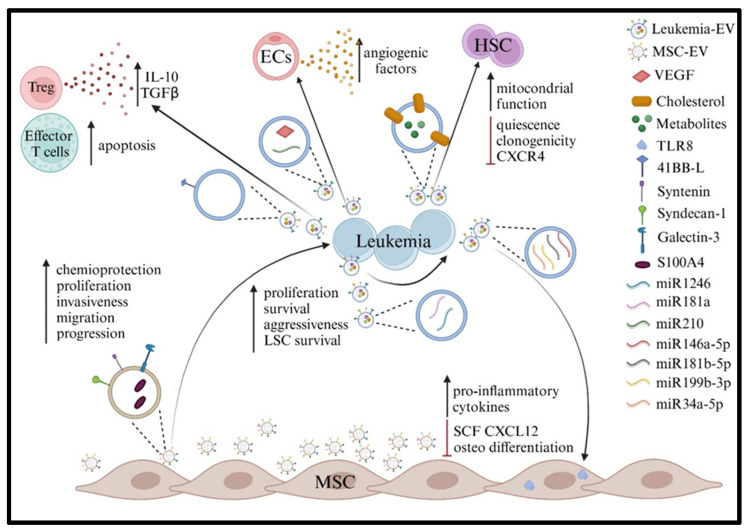
Created in BioRender. D’Amico, G. (2025); https://BioRender.com/na8vkbw.

**Figure 2 ijms-26-05380-f002:**
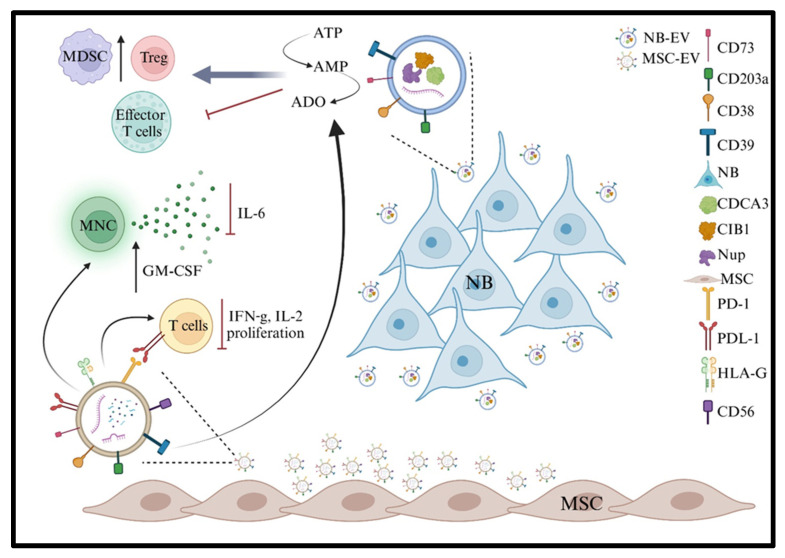
Created in BioRender. Airoldi, I. (2025); https://BioRender.com/45fho1i.

**Figure 3 ijms-26-05380-f003:**
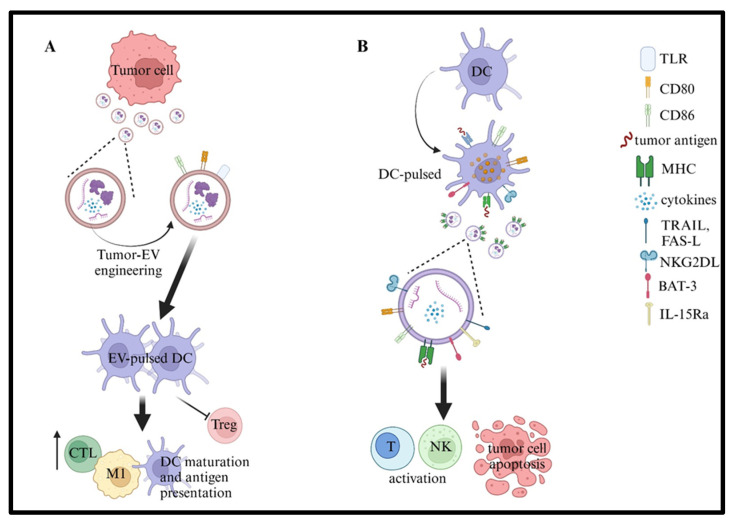
Created in BioRender. Airoldi, I. (2025); https://BioRender.com/mplge2w.

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
