# Peer review of "Dysregulation of the Bone Marrow Microenvironment in Pediatric Tumors: The Role of Extracellular Vesicles in Acute Leukemias and Neuroblastoma"

_ijms, 2025, doi:10.3390/ijms26115380_

Round 1
Reviewer 1 Report
Comments and Suggestions for Authors
Could you please add more figures, otherwise a very well structured review!
Comments on the Quality of English LanguageEnglish fine!
Author Response
We thank the Reviewer for His/Her positive comment.
Reviewer 2 Report
Comments and Suggestions for Authors
The manuscript „Dysregulation of the bone marrow microenvironment in pediatric tumors: the role of extracellular vesicles in acute leukemias and neuroblastoma“ from Giovanna D’amico and colleagues addresses an important topic—the role of extracellular vesicles (EVs) in the dysregulation of the bone marrow microenvironment in pediatric tumors. The review offers valuable insights and brings together relevant findings from the literature. However, several aspects need to be improved to enhance clarity and impact. In its current form, the manuscript lacks a cohesive narrative structure and does not fully meet the needs of both biological and medical readers. The flow between sections is uneven, and transitions are abrupt. A more consistent and logical structure is recommended to guide the reader through the various aspects of the topic. Please consider adding introductory and bridging sentences between sections to clarify why certain concepts follow others and how they are connected.
The manuscript would benefit from a clearer clinical framing. Introducing key statistics on the prevalencevand prognosis for pediatric acute leukemias and neuroblastoma would help underscore the importance of studying the bone marrow microenvironment in these diseases. The connection between the selected tumors and the bone marrow microenvironment needs to be better articulated. For an experimental bioscientist, like me, these two entities are nor really connected. This context is essential for readers from clinical backgrounds to appreciate the translational relevance of EV research. For example, in neuroblastoma, which is not primarily a bone marrow-originating tumor, the relevance of BM involvement and EV-mediated crosstalk should be more explicitly explained.
While the review references the involvement of EVs, it would benefit from a clearer and more detailed discussion of the known or hypothesized mechanisms through which EVs contribute to niche remodeling, immune evasion, or therapy resistance.
The manuscript currently lacks a concluding section. Adding a summary and forward-looking perspective is essential. Other points to include:
- How EV research may inform future diagnostic or therapeutic strategies
- The potential for EVs as biomarkers in pediatric oncology
- The need for further research into cell-type-specific EV signaling within the bone marrow
- Some typographical and grammatical errors are present and should be corrected throughout the manuscript.
The topic is timely and promising, but the maunscript requires substantial revision to improve structure, clarity, and relevance. With careful restructuring and enhanced mechanistic and clinical framing, it could become a valuable contribution to the field.
Author Response
Comment 1. The manuscript „Dysregulation of the bone marrow microenvironment in pediatric tumors: the role of extracellular vesicles in acute leukemias and neuroblastoma“ from Giovanna D’amico and colleagues addresses an important topic—the role of extracellular vesicles (EVs) in the dysregulation of the bone marrow microenvironment in pediatric tumors. The review offers valuable insights and brings together relevant findings from the literature. However, several aspects need to be improved to enhance clarity and impact. In its current form, the manuscript lacks a cohesive narrative structure and does not fully meet the needs of both biological and medical readers. The flow between sections is uneven, and transitions are abrupt. A more consistent and logical structure is recommended to guide the reader through the various aspects of the topic. Please consider adding introductory and bridging sentences between sections to clarify why certain concepts follow others and how they are connected.
Response 1. We agree with the Reviewer and, accordingly, we have added bridging sentences between sections as in paragraph 1 lines 74-79, paragraph 2 lines 111-113, paragraph 4 lines 223-226, and paragraph 8 lines 390-392.
Comment 2. The manuscript would benefit from a clearer clinical framing. Introducing key statistics on the prevalencev and prognosis for pediatric acute leukemias and neuroblastoma would help underscore the importance of studying the bone marrow microenvironment in these diseases. The connection between the selected tumors and the bone marrow microenvironment needs to be better articulated. For an experimental bioscientist, like me, these two entities are nor really connected. This context is essential for readers from clinical backgrounds to appreciate the translational relevance of EV research. For example, in neuroblastoma, which is not primarily a bone marrow-originating tumor, the relevance of BM involvement and EV-mediated crosstalk should be more explicitly explained.
Response 2. We thank the Reviewer for this comment. In order to cope with this criticism, we first included “the key statistics on the prevalence and prognosis for pediatric acute leukemias and neuroblastoma” in paragraph 3 lines 120-129, and paragraph 4 lines 191-192 and 209-212, as requested. The appropriate new references 42 and 64 have been added in the text. Next, the “The connection between the selected tumors and the bone marrow microenvironment” is established by the knowledge that acute leukemias arise from BM progenitors and approximately 50% of neuroblastoma patients show BM metastasis at diagnosis and present very poor prognosis. This concept is now better defined in the abstract (line 22-23), introduction (lines 74-79), paragraph 3 lines 164-168, paragraph 4 lines 223-226 and in the new section of the concluding remarks. Next, the “The connection between the selected tumors and the bone marrow microenvironment” is established by the knowledge that acute leukemias arise from BM progenitors and approximately 50% of neuroblastoma patients show BM metastasis at diagnosis and present very poor prognosis. This concept is now better defined in the abstract (line 22-23), introduction (lines 74-79), paragraph 3 lines 163-167, paragraph 4 lines 223-226 and in the new section of the concluding remarks.
Comment 3. While the review references the involvement of EVs, it would benefit from a clearer and more detailed discussion of the known or hypothesized mechanisms through which EVs contribute to niche remodeling, immune evasion, or therapy resistance.
The manuscript currently lacks a concluding section. Adding a summary and forward-looking perspective is essential. Other points to include:
- How EV research may inform future diagnostic or therapeutic strategies
- The potential for EVs as biomarkers in pediatric oncology
- The need for further research into cell-type-specific EV signaling within the bone marrow
- Some typographical and grammatical errors are present and should be corrected throughout the manuscript.
Response 3. We have now added in the review a new paragraph 11 named “concluding remarks” (lines 502-519) that includes these points. Furthermore, we have double checked the errors and corrected them.
Comment 4. The topic is timely and promising, but the maunscript requires substantial revision to improve structure, clarity, and relevance. With careful restructuring and enhanced mechanistic and clinical framing, it could become a valuable contribution to the field.
Response 4. We thank the Reviewer for His/Her comments and suggestions that greatly increased the quality of the work. We are now confident to have satisfied the requests of the Reviewer.
Reviewer 3 Report
Comments and Suggestions for Authors
This MS provides an insightful overview of the role of EVs in pediatric tumors, especially acute leukemia and neuroblastoma. It summarizes the intricate interplay between cancer-derived EVs and the bone marrow microenvironment. It also highlights the role of cancer-derived EVs in MDR and poor survival for some patients. The paper is well-structured, scientifically rigorous, and contributes meaningfully to our understanding of tumor progression in children. Overall, it is a significant and timely addition that summarizes the role of EVs in the field of pediatric oncology management..
Author Response
Comment 1. This MS provides an insightful overview of the role of EVs in pediatric tumors, especially acute leukemia and neuroblastoma. It summarizes the intricate interplay between cancer-derived EVs and the bone marrow microenvironment. It also highlights the role of cancer-derived EVs in MDR and poor survival for some patients. The paper is well-structured, scientifically rigorous, and contributes meaningfully to our understanding of tumor progression in children. Overall, it is a significant and timely addition that summarizes the role of EVs in the field of pediatric oncology management.
Response 1. We are very grateful to the Reviewer for His/Her positive comment.
Reviewer 4 Report
Comments and Suggestions for Authors
The authors have provided a review detailing the role of extracellular vesicles (EVs) in ALL/AML and neuroblastoma (NB). All of these are devastating pediatric malignancies where additional research is needed to further improve outcomes. The role of EVs in cancer and cancer therapy has greatly expanded in recent years and this review provides a good example of many of the new findings. There are a few missing pieces, however, that could be included to further strengthen the work and provide a broader impact for the paper. These are outlined below.
- In section 2 (Role of EVs in immune-modulation) there is much important information provided detailing the role of EVs in immune modulation, but very few references given to support. Please include additional references, particularly in paragraphs 2 and 3.
- In section 3 (ALL & AML) the authors describe that there are various molecular changes associated with the pathogenesis of these diseases. Could the authors kindly add examples and supporting references?
- In section 4 (NB) the authors describe NB but do not describe any associated genetic alterations (aka chromosomal alterations or amplifications other than MYCN) associated with the disease. Could they kindly add this additional information and associated references?
- In section 5 (BM niche in physiology), could the authors provide more information and references to support the described findings in lines 194-197?
- In section 6 (BM environment in ALL/AML), could the authors kindly provide additional information and/or examples of some of the cancer-derived signals modifying the BM environment as mentioned in lines 200-202?
- In section 8 (BM in NB), could the authors kindly provide some additional references to support the role of NB cells in BM niche modulation? For example, the authors describe how MCS enhance NB growth, reduce osteoblast differentiation, and increase osteoclast function but no references are provided. Also, references for RANKL signaling are needed.
- The manuscript could benefit from the inclusion of a Discussion or Conclusion section to summarize the work provided and give future insights.
- Minor point throughout: Could the authors double-check that all gene/protein names are described in full prior to abbreviations being used throughout the work.
Author Response
Comment 1. In section 2 (Role of EVs in immune-modulation) there is much important information provided detailing the role of EVs in immune modulation, but very few references given to support. Please include additional references, particularly in paragraphs 2 and 3.
Response 1. We thank the Reviewer for His/Her suggestion. Accordingly, we have now included new references in paragraph 2 (14-16 and 18-39) and paragraph 3 (41-47, 52-55 and 58-60).
Comment 2. In section 3 (ALL & AML) the authors describe that there are various molecular changes associated with the pathogenesis of these diseases. Could the authors kindly add examples and supporting references?
Response 2. We agree with this comment and we have extended data regarding this important point. Thus, we have added a paragraph in section 3 describing the most frequent genetic alteration in B-ALL (lines 133-146) with the corresponding new references 46-47.
Comment 3. In section 4 (NB) the authors describe NB but do not describe any associated genetic alterations (aka chromosomal alterations or amplifications other than MYCN) associated with the disease. Could they kindly add this additional information and associated references?
Response 3. Thanks to this comment, we have included in the manuscript additional genetic alterations associated with NB at paragraph 4 lines 200-212 and the appropriate new references 63-64.
Comment 4. In section 5 (BM niche in physiology), could the authors provide more information and references to support the described findings in lines 194-197?
Response 4. As requested, we have now added references 37, 73, 76-78.
Comment 5. In section 6 (BM environment in ALL/AML), could the authors kindly provide additional information and/or examples of some of the cancer-derived signals modifying the BM environment as mentioned in lines 200-202?
Response 5. We thank the Reviewer for His/Her suggestion Accordingly, we have expanded the description of leukemia-derived signals that can modify the surrounding microenvironment in both ALL and AML, as now detailed in Section 6 (lines 263-286). We have also added additional references to support the revised text (references 79 and 83-89).
Comment 6. In section 8 (BM in NB), could the authors kindly provide some additional references to support the role of NB cells in BM niche modulation? For example, the authors describe how MCS enhance NB growth, reduce osteoblast differentiation, and increase osteoclast function but no references are provided. Also, references for RANKL signaling are needed.
Response 6. We thank the Reviewer for this comment. Accordingly, we have included in section 8 a new sentence a lines 409-413 with references 116 and 118, as well as new reference 117 for RANKL.
Comment 7. The manuscript could benefit from the inclusion of a Discussion or Conclusion section to summarize the work provided and give future insights.
Response 7. Thanks to this request, we have now included the final paragraph “concluding remarks”.
Comment 8. Minor point throughout: Could the authors double-check that all gene/protein names are described in full prior to abbreviations being used throughout the work.
Response 8. We are sorry for this point. We have checked the gene/protein names and included full names lost before the abbreviations throughout the manuscript.
Round 2
Reviewer 2 Report
Comments and Suggestions for Authors
The authors have adressed many concerns and the article has improved a lot, so I have no further objections.
Reviewer 4 Report
Comments and Suggestions for Authors
The authors have resolved all of my earlier suggestions associated with the previous version of the manuscript. I hope they found my suggestions useful.